# ONE-PIXEL SHORTCUT: ON THE LEARNING PREFERENCE OF DEEP NEURAL NETWORKS

**Shutong Wu** [*1]**, Sizhe Chen** [*1]**, Cihang Xie**[2] **& Xiaolin Huang** [1]
[1]Department of Automation, Shanghai Jiao Tong University
[2]Computer Science and Engineering, University of California, Santa Cruz

## ABSTRACT

Unlearnable examples (ULEs) aim to protect data from unauthorized usage for training DNNs. Existing work adds $\ell_\infty$-bounded perturbations to the original sample so that the trained model generalizes poorly. Such perturbations, however, are easy to eliminate by adversarial training and data augmentations. In this paper, we resolve this problem from a novel perspective by perturbing only one pixel in each image. Interestingly, such a small modification could effectively degrade model accuracy to almost an untrained counterpart. Moreover, our produced *One-Pixel Shortcut (OPS)* could not be erased by adversarial training and strong augmentations. To generate OPS, we perturb in-class images at the same position to the same target value that could mostly and stably deviate from all the original images. Since such generation is only based on images, OPS needs significantly less computational cost than the previous methods using DNN generators. Based on OPS, we introduce an unlearnable dataset called CIFAR-10-S, which is indistinguishable from CIFAR-10 by humans but induces the trained model to extremely low accuracy. Even under adversarial training, a ResNet-18 trained on CIFAR-10-S has only 10.61% accuracy, compared to 83.02% by the existing error-minimizing method.

## 1 INTRODUCTION

Deep neural networks (DNNs) have successfully promoted the computer vision field in the past decade. As DNNs are scaling up unprecedentedly (Brock et al., 2018; Huang et al., 2019; Riquelme et al., 2021; Zhang et al., 2022), data becomes increasingly vital. For example, ImageNet (Russakovsky et al., 2015) fostered the development of AlexNet (Krizhevsky et al., 2017). Besides, people or organizations also collect online data to train DNNs, *e.g.*, IG-3.5B-17k (Mahajan et al., 2018) and JFT-300M (Sun et al., 2017). This practice, however, raises the privacy concerns of Internet users. In this concern, researchers have made substantial efforts to protect personal data from abuse in model learning without affecting user experience (Feng et al., 2019; Huang et al., 2020a; Fowl et al., 2021; Yuan & Wu, 2021; Yu et al., 2021). Among those proposed methods, unlearnable examples (ULEs) (Huang et al., 2020a) take a great step to inject original images with protective but imperceptible perturbations from bi-level error minimization (EM). DNNs trained on ULEs generalize very poorly on normal images. However, such perturbations could be completely canceled out by adversarial training, which fails the protection, limiting the practicality of ULEs.

We view the data protection problem from the perspective of shortcut learning (Geirhos et al., 2020), which shows that DNN training is "lazy" (Chizat et al., 2019; Caron & Chrétien, 2020), *i.e.*, converges to the solution with the minimum norm when optimized by gradient descent (Wilson et al., 2017; Shah et al., 2018; Zhang et al., 2021). In this case, a DNN would rely on every accessible feature to minimize the training loss, no matter whether it is semantic or not (Ilyas et al., 2019; Geirhos et al., 2018; Baker et al., 2018). Thus, DNNs tend to ignore semantic features if there are other easy-to-learn shortcuts that are sufficient for distinguishing examples from different classes. Such shortcuts exist naturally or manually. In data collection, *e.g.*, cows may mostly appear with grasslands, misleading DNN to predict cows by large-area green, because the color is easier to learn than those semantic features and also sufficient to correctly classify images of cows during training. Such natural shortcuts

---

[*]equal contribution; correspondence to Xiaolin Huang (xiaolinhuang@sjtu.edu.cn).
[†]code available at `https://github.com/cychomatica/One-Pixel-Shotcut`.

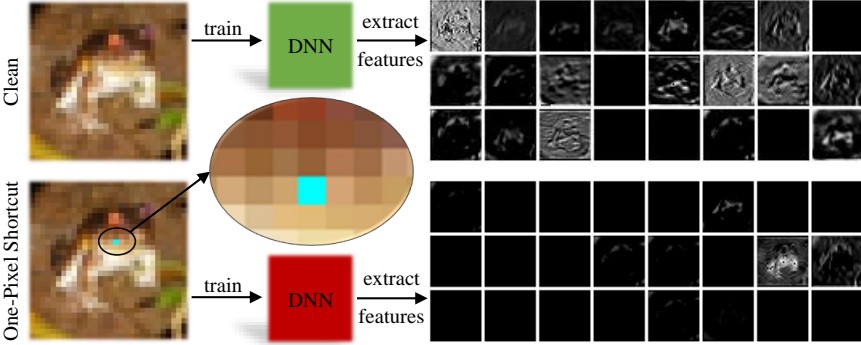

Figure 1: Effect of One-Pixel Shortcut. We visualize the features (after the first convolution) of the ResNet-18 (He et al., 2016) models trained by clean and OPS samples. Even at such a shallow layer, the DNN trained on OPS extracts much fewer semantic features and is less activated.

have been illustrated in detail in datasets such as ImageNet-A (Hendrycks et al., 2021) and ObjectNet (Barbu et al., 2019). Besides, shortcuts could also be manually crafted. For instance, EM-based ULEs (Huang et al., 2020a) mislead DNNs to learn features belonging to the perturbations, which falls in the category of shortcut learning (Yu et al., 2021).

In this paper, we are surprised to find that shortcuts could be so small in the area that *it can even be simply instantiated as a single pixel*. By perturbing a pixel of each training sample, our method, namely One-Pixel Shortcut (OPS), degrades the model accuracy on clean data to almost an untrained counterpart. Moreover, our generated unbounded small noise could not be erased by adversarial training (Madry et al., 2018), which is effective in mitigating existing ULEs (Huang et al., 2020a). To make the specific pixel stand out in view of DNNs, OPS perturbs in-class images at the same position to the same target value that, if changed to a boundary value, could mostly and stably deviate from all original images. Specifically, the difference between the perturbed pixel and the original one in all in-class images should be large with low variance. Since such generation is only based on images, OPS needs significantly less computational cost than the previous methods based on DNN generators. We evaluate OPS and its counterparts in 6 architectures, 6 model sizes, 8 training strategies on CIFAR-10 (Krizhevsky et al., 2009) and ImageNet (Russakovsky et al., 2015) subset, and find that OPS is always superior in degrading model's testing accuracy than EM ULEs. In this regard, we introduce a new unlearnable dataset named CIFAR-10-S, which combines the EM and OPS to craft stronger imperceptible ULEs. Even under adversarial training, a ResNet-18 (He et al., 2016) trained on CIFAR-10-S has 10.61% test accuracy, compared to 83.02% by the existing error-minimizing method. Different from the existing datasets like ImageNet-A (Hendrycks et al., 2021) or ObjectNet (Barbu et al., 2019), which place objects into special environments to remove shortcuts, CIFAR-10-S injects shortcuts to evaluate the model's resistance to them. Altogether, our contributions are included as follows:

- We analyze unlearnable examples from the perspective of shortcut learning, and demonstrate that a strong shortcut for DNNs could be as small as a single pixel.
- We propose a novel data protection method named One-Pixel Shortcut (OPS), which perturbs in-class images in the pixel that could mostly and stably deviate from the original images. OPS is a model-free method that is significantly faster than previous work.
- We extensively evaluate OPS on various models and training strategies, and find it outperforms baselines by a large margin in the ability to degrade DNN training. Besides, we introduce CIFAR-10-S to assess the model's ability to learn essential semantic features.

## 2  RELATED WORK

### 2.1  ADVERSARIAL ATTACK AND DATA POISONING

Adversarial examples are perturbed by small perturbations, which are indistinguishable from the original examples by humans but can make DNNs give wrong predictions (Szegedy et al., 2014).

Many different adversarial attacks are proposed in recent years. Generally, most adversarial attacks aim to perturb the whole image with a constrained intensity (usually bounded by $\ell_p$ norm), *e.g.*, PGD (Madry et al., 2018), C&W (Carlini & Wagner, 2017) and Autoattack (Croce & Hein, 2020). Besides, there are also other methods that only perturb a small part of an image (Croce & Hein (2019); Dong et al. (2020)) or even a single pixel (Su et al. (2019)). The existence of adversarial examples and their transferability (Chen et al. (2022)) indicates that DNNs do not sufficiently learn critical semantic information as we wish, but more or less depend on some non-robust features.

Data poisoning aims to modify the training data in order to affect the performance of models. Usually, the poisoned examples are notably modified and take only a part of the whole dataset (Yang et al., 2017; Koh & Liang, 2017). But those methods cannot degrade the performance of models to a low enough level, and the poisoned examples are easily distinguishable. Recently researchers have paid great attention to imperceptible poisoning which modifies examples slightly and does not damage their semantic information (Huang et al., 2020a; Fowl et al., 2021; Huang et al., 2020b; Doan et al., 2021; Geiping et al., 2020; Chen et al., 2023). Fowl et al. (2021) uses adversarial perturbations which contain information of wrong labels to poison the training data, which is equivalent to random label fitting. On the contrary, Huang et al. (2020a) attack the training examples inversely, *i.e.*, using error-minimizing perturbation, to craft unlearnable examples.

## 2.2 SHORTCUT LEARNING

Recently, researches on deep neural networks indicate that under the sufficiency of correct classification, DNNs tend to learn easier features instead of semantic features which make the object itself. To be more specific, for example, the same object in different environments will get different predictions, which means the DNN overly relies on features that do not belong to the object (Beery et al., 2018). Geirhos et al. (2020) investigate this phenomenon in different fields of deep learning and explain why shortcuts exist and how to understand them. Lapuschkin et al. (2019) also observe this problem and attribute it to the unsuitable performance evaluation metrics that we generally use. The existence of natural adversarial examples (Hendrycks et al., 2021) also indicates that DNNs do not sufficiently learn the real semantic information during training. Instead, they may learn to use the background or texture of an image to predict. Unlearnable examples (ULEs) (Huang et al., 2020a), which are manually crafted by error-minimizing noises and able to lead the models trained on them to obtain terrible generalization on test data, are believed to be some kind of shortcut that provides some textures that are easy to learn (Yu et al., 2021). Generally, if we get enough data, the interconnection of different features will be enhanced so that those shortcuts may not be sufficient for classification tasks, *i.e.*, the model will have to use more complicated composed features in order to minimize the risk. However, when the data we collect obtains some specific bias (*e.g.*, similar backgrounds), shortcut learning will not be mitigated effectively.

## 2.3 DATA AUGMENTATION

Data augmentation aims to enhance the generalization ability of models. This is usually implemented by applying some transformations to the training data, *e.g.*, random stretching, random cropping, or color changing. Nowadays different kinds of data augmentation policies (Zhang et al., 2018; DeVries & Taylor, 2017; Cubuk et al., 2019; 2020) are proven to effectively boost the generalization ability of DNNs. Sometimes adversarial training Madry et al. (2018); Li et al. (2022) is also regarded as a kind of data augmentation (Shorten & Khoshgoftaar, 2019; Xie et al., 2020). Dao et al. (2019) deem data augmentation as a kind of data-dependent regularization term. Since data augmentations are believed to improve the generalization ability of DNNs, we use different augmentations to evaluate the effectiveness of different data protection methods.

# 3 ONE-PIXEL SHORTCUT

## 3.1 PRELIMINARIES

Unlearnable examples (ULEs) are a data protection method created by error-minimizing (EM) noises (Huang et al., 2020a). Models trained on examples that are perturbed by those noises will get almost zero training error, but perform like random guessing on clean test data. Due to the imperceptibility

of the noise, this method can prevent the abuse of data by some unauthorized users who attempt to train deep models for improper purposes, without affecting normal usage. This bi-level problem can be solved by optimizing the inner minimization and the outer minimization alternately. It is proved that the perturbations belonging to the same class are well clustered and linearly separable (Yu et al., 2021). Thus, EM provides easy-to-learn features which are closely interconnected with labels.

We design a shuffling experiment to demonstrate that DNNs learn the shortcuts instead of images if trained on data that have shortcuts. Denote $\hat{\mathcal{D}} = \{(x_i + \delta_i, y_i)\}$ as the perturbed training set, where $\delta_i$ is the perturbation associated with the $i$-th example. After shuffling, the training set becomes $\hat{\mathcal{D}}' = \{(x_j + \delta_i, y_i)\}$. We are curious about how the DNNs trained on different data (with or without shortcuts) would predict the shuffled perturbed training set. As shown in Table 1, the DNN trained on clean data performs like a random guess on shuffled perturbed data while keeping high accuracy on unshuffled data, indicating that it successfully learns the representations from images $\{x_i\}$ rather than giving predictions according to the perturbations $\{\delta_i\}$ from EM or OPS. In contrast, the DNN trained on EM training data tends to memorize a significant proportion of perturbations, because it has 48.97% accuracy on the shuffled EM training data. Moreover, OPS training data produces a DNN with 72.43% accuracy on the shuffled set, reflecting it forcing the DNN to learn the perturbations to a greater extent. Our study illustrates the learning characteristics of DNNs trained with or without shortcuts, and also shows that OPS is a more effective shortcut than EM.

Table 1: The testing accuracy of ResNet-18 models trained on unshuffled and shuffled data.

| Testing / Training | Clean | EM | OPS |
|---|---|---|---|
| Clean | 99.96 | 25.16 | 15.48 |
| EM | 94.00 | 99.99 | 15.54 |
| Shuffled EM | 9.97 | **48.97** | 10.15 |
| OPS | 99.52 | 25.07 | 99.97 |
| Shuffled OPS | 9.82 | 10.35 | **72.43** |

## 3.2 FOOLING DNN TRAINING BY A PIXEL

Following the discussion above, for the purpose of data protection, we need to craft shortcuts that are easy enough to learn and thus fool the network training. According to previous studies, shortcuts can come from background environments that naturally exist inside our datasets (Beery et al., 2018), or be manually crafted like EM (Huang et al., 2020a). Unlike those shortcuts which might occupy the whole image or a notable part, we investigate how a single pixel, which is the minimum unit of digital images, can affect the learning process of deep neural networks. Thus, we propose One-Pixel Shortcut (OPS), which modifies only a single pixel of each image. Images belonging to the same category are perturbed at the same position, which means the perturbed pixel is interconnected with the category label. Although so minuscule, it is efficient enough to fool the training of deep learning models. We use a heuristic but effective method to generate perturbations for images belonging to each category. We search the position and value of the pixel which can result in the most significant change for the whole category. Denoting $\mathcal{D}$ as the clean dataset and $\mathcal{D}_k$ as the clean subset containing all the examples of class $k$, the problem can be formulated as:

$$\arg\max_{\sigma_k, \xi_k} \mathbb{E}_{(x,y) \in \mathcal{D}_k} \left[ \mathcal{G}_k \left( x, \sigma_k, \xi_k \right) \right] \qquad \text{s.t.} \quad \|\sigma_k\|_0 = 1, \sum_{i,j} \sigma_k(i,j) = 1, \tag{1}$$

where $\sigma_k \in \mathbb{R}^{H \times W}$ represents the perturbed position mask and $\sigma_k(i,j)$ is the element at the $i$-th row and $j$-th column, $\xi_k \in \mathbb{R}^C$ stands for the perturbed target color ($C = 3$ for RGB images), and $\mathcal{G}$ is the objective function. Since the optimization above is an NP-hard problem, we cannot solve it directly. Thus we constrain the feasible region to a limited discrete searching space, where we search the boundary value of each color channel, *i.e.*,$\xi_k \in \{0, 1\}^3$, at every point of an image. Specifically, for CIFAR-10 images, the discrete searching space will contain $32 \times 32 \times 2^3 = 8192$ elements. To

ensure that the pixel is stably perturbed, we also hope that the variance of the difference between them is small. Accordingly, we design the objective function $\mathcal{G}_k$ for class $k$ as:

$$\mathcal{G}_k = \frac{\mathbb{E}_{(x,y)\in\mathcal{D}_k}\left(\sum_{j=1}^{C}|\|x_j \cdot \sigma_k\|_F - \xi_{kj}|\right)}{\mathrm{Var}_{(x,y)\in\mathcal{D}_k}\left(\sum_{j=1}^{C}|\|x_j \cdot \sigma_k\|_F - \xi_{kj}|\right)} \tag{2}$$

where $x_i \in \mathbb{R}^{H\times W}$ denotes the $i$-th channel of $x$, and $\xi_{kj} \in \mathbb{R}$ is the $i$-th channel of $\xi_k$. After solving the position map and color, we get perturbation $\delta$ for each example $(x, y)$ as:

$$\delta = [\xi_{y1}\sigma_y - x_1 \cdot \sigma_y, \xi_{y2}\sigma_y - x_2 \cdot \sigma_y, \dots, \xi_{yC}\sigma_y - x_C \cdot \sigma_y]^\top \tag{3}$$

Details can be found in Algorithm 1. The resulting One-Pixel Shortcut is illustrated in Figure 2.

---

**Algorithm 1** Model-Free Searching for One-Pixel Shortcut

---

**Input:** Clean dataset $\mathcal{D} = \mathcal{D}_1 \bigcup \cdots \bigcup \mathcal{D}_M$
**Output:** One-Pixel Shortcut dataset $\hat{\mathcal{D}} = \hat{\mathcal{D}}_1 \bigcup \cdots \bigcup \hat{\mathcal{D}}_M$
 1: **for** $k = 1, 2, 3, ..., M$ **do**
 2:    solve Eq.1 and Eq.2 to get $\sigma_k$ and $\xi_k$         # *calculate the best perturbed point for class k*
 3:    **for** each $x \in \mathcal{D}_k$ **do**
 4:       **for** $i = 1, 2, 3$ **do**
 5:          $\hat{x}_i = x_i \cdot (I - \sigma_k) + \xi_{ki} \cdot \sigma_k$    # *modify the optimal pixel for every image in class k*
 6:       **end for**
 7:    **end for**
 8:    $\hat{\mathcal{D}}_k = \{\hat{x}\}$
 9: **end for**
10: **return** $\hat{\mathcal{D}} = \hat{\mathcal{D}}_1 \bigcup \cdots \bigcup \hat{\mathcal{D}}_M$

---

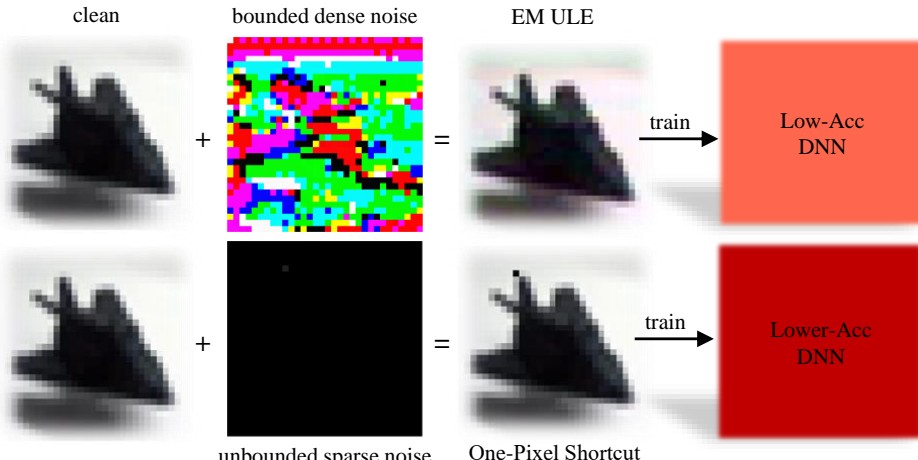

Figure 2: The effects of Error-Minimizing noise and One-Pixel Shortcut. Models trained on either EM or OPS perturbed data get abnormally low accuracy on clean test data. To ensure imperceptibility, EM adds $\ell_p$ bounded noise to the whole image, while OPS applies unbounded but sparse perturbation which only affects a small part of the image.

## 3.3 PROPERTIES OF ONE-PIXEL SHORTCUT

Since we all empirically believe that convolutional networks tend to capture textures (Hermann et al., 2020) or shapes (Geirhos et al., 2018; Zhang & Zhu, 2019), it is surprising that convolutional networks can be affected so severely by just one pixel. As illustrated by Figure 1, the network indeed

tends to learn those less complicated nonsemantic features brought by One-Pixel Shortcut. Besides convolutional networks, we observe that compact vision transformers (Hassani et al., 2021) are also attracted by One-Pixel Shortcut and ignore other semantic features. This indicates that shortcuts are not particularly learned by some specific architecture. We also visualize the loss landscape of ResNet-18 trained on clean CIFAR-10 data and One-Pixel Shortcut data. Illustrated as Figure 3, while trained on OPS data, the loss surface is much flatter, which means that these minima found by the network are more difficult to escape. Even if we use a ResNet-18 pretrained on clean CIFAR-10 and then fine-tune it on the OPS data, the network will still fall into this badly generalized minima.

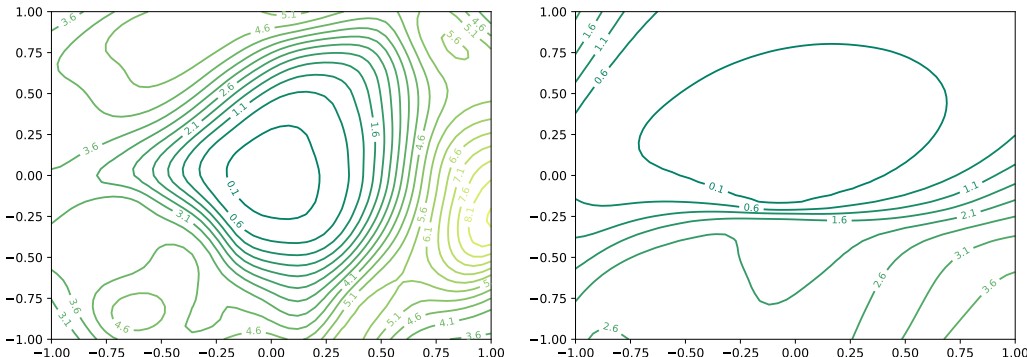

Figure 3: Loss landscape visualization (Li et al., 2018) of ResNet-18 trained on clean CIFAR-10 data and our One-Pixel Shortcut data. The landscape of THE OPS-trained model is flatter, making it minima harder to escape.

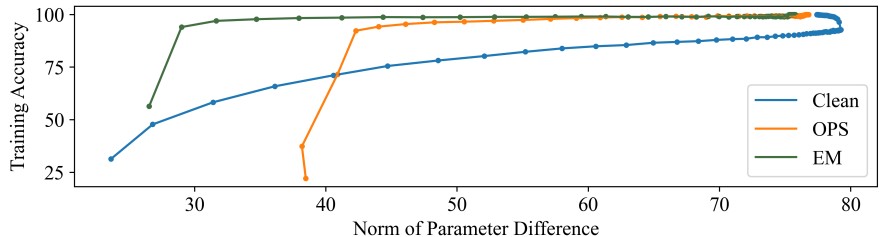

Figure 4: Training accuracy and the Frobenius norm of parameter difference (*i.e.*, $\|\theta - \theta_0\|_F$) when ResNet-18 are trained on different training data. While training on EM or OPS data, the network tends to find an optimum closer to the initialized point. This is consistent with the view of shortcut learning.

In addition, we record the trajectories of training accuracy and the Frobenius norm of parameter difference, $\|\theta - \theta_0\|_F$, which can reflect the magnitude of network parameter change. Here $\theta$ and $\theta_0$ respectively indicate the parameters after training and at the initialized point. We draw the relation curve between training accuracy and $\|\theta - \theta_0\|_F$, which can be found in Figure 4. When training accuracy rises up to 90% for the first time, the model trained on OPS data has a much smaller $\|\theta - \theta_0\|_F$ than that trained on clean data, which indicates that the OPS-trained model gets stuck in an optimum closer to the initialization. It has been widely known that overparameterized DNNs optimized by gradient descent will converge to the solution that is close to the initialization, *i.e.* with the minimum norm of parameter difference (Wilson et al., 2017; Shah et al., 2018; Li & Liang, 2018; Zhang et al., 2021). Since OPS only perturbs a single pixel, the original representations of images are not damaged, and the model trained on clean data can still keep great performance on OPS data, which indicates that the well-generalized solution far from the initialization still exists but is not reached due to the tendency for a close solution. The close solution is believed to obtain better generalization ability. Nevertheless, this argument is true only under the assumption that the training data and test data are from the exact same distribution and have the exact same features. The existence of OPS forces the model to converge to an optimum where the model generalizes well on OPS features, which are not contained in test data. From our experiment results in Table 2, OPS can degrade test accuracy to a lower level. This is because EM requires a generator model, and thus may

contain features more or less depending on it, which constrains the effectiveness of other models. On the other hand, OPS is a universal model-free method, and the shortcuts are crafted based on the inherent learning preference of DNNs.

# 4 EXPERIMENTS

## 4.1 SETTING

Our experiments are implemented on CIFAR-10 and ImageNet subset, using 4 NVIDIA RTX 2080Ti GPUs. We investigate how the One-Pixel Shortcut can affect the training of different models (including different architectures and different capacities). We evaluate our method on different models including convolutional networks (He et al., 2016; Zagoruyko & Komodakis, 2016; Huang et al., 2017) and the recently proposed compact vision transformers (Hassani et al., 2021). For all the convolutional networks, we use an SGD optimizer with a learning rate set to $0.1$, momentum set to $0.9$, and weight decay set to $5e-4$. For all the compact vision transformers, we use AdamW optimizer with $\beta_1 = 0.9$, $\beta_2 = 0.999$, learning rate set to $5e-4$, and weight decay set to $3e-2$. Batch size is set to $128$ for all the models except WideResNet-28-10, where it is set to $64$.

Table 2: Clean test accuracy on CIFAR-10 models of different architectures.

| Model | Training Data | | |
|---|---|---|---|
| | Clean | EM | OPS (Ours) |
| LeNet-5 | 70.27 | 26.98 | 22.19 |
| CVT-7-4 | 87.46 | 27.60 | 18.21 |
| CCT-7-3×1 | 88.98 | 27.06 | 17.95 |
| DenseNet-121 | 94.10 | 23.72 | 11.45 |
| ResNet-18 | 94.01 | 19.58 | 15.56 |
| WideResNet-28-10 | 96.08 | 23.96 | 12.76 |

Besides, We also try different training strategies including adversarial training (Madry et al., 2018) and different data augmentations (Mixup (Zhang et al., 2018), Cutout (DeVries & Taylor, 2017) and RandAugment (Cubuk et al., 2020)). For adversarial training, we use a 10-step PGD attack, setting step size to $2/255$ and $\ell_\infty$ bound to $8/255$. For Mixup, Cutout, and RandAugment, we use the default settings from their original papers. All the models are trained for 200 epochs with a multi-step learning rate schedule, and the training accuracy of each model is guaranteed to reach near 100%.

We additionally tested our method on the ImageNet subset (the first 100 classes). We center-crop all the images to $224 \times 224$ and train common DNNs with results in Table 4. We adopt an initial learning rate of 0.1 with a multi-step learning rate scheduler and train models for 200 epochs. Our One-Pixel Shortcut can still be effective in protecting large-scale datasets. The networks trained on OPS data will get much lower clean test accuracy than those trained on clean data.

Table 3: WideResNets of different capacities trained on One-Pixel Shortcut data. Test Acc. stands for accuracy on the clean CIFAR-10 testset.

| | WRN-28-1 | WRN-28-2 | WRN-28-4 | WRN-28-8 | WRN-28-16 | WRN-28-20 |
|---|---|---|---|---|---|---|
| Size | 0.37M | 1.47M | 5.85M | 23.36M | 93.35M | 145.84M |
| Test Acc. | 21.20 | 14.74 | 21.75 | 18.01 | 15.36 | 18.04 |

## 4.2 EFFECTIVENESS ON DIFFERENT MODELS

We train different convolutional networks and vision transformers on the One-Pixel Shortcut CIFAR-10 training set, and evaluate their performance on the unmodified CIFAR-10 test set. Details are shown in Table 2. Every model reaches a very high training accuracy after only several epochs, which is much faster than training on clean data. Meanwhile, they all get really low test accuracy (about 15%) on clean test data, indicating that they do not generalize at all.

Table 4: Clean test accuracy on ImageNet models of different architectures.

| Model | Training Data | |
|---|---|---|
| | Clean | OPS (Ours) |
| ResNet-18 | 76.18 | 9.68 |
| ResNet-50 | 64.26 | 10.38 |
| DenseNet-121 | 75.14 | 11.48 |

Although the perturbed image looks virtually the same as the original image, and all the models get near 100% training accuracy quickly, they do not capture any semantic information but just

the pixels we modify in the images. We also train models on EM training set, which is generated by a ResNet-18 using the official implementation of Huang et al. (2020a). The $\ell_\infty$ bound of EM noises is set to $8/255$. The generation of OPS costs only about 30 seconds, which is much faster than EM costing about half an hour. For different networks, OPS can degrade their test accuracy to a lower level than EM. EM works the best on ResNet-18 (19.58% test accuracy), which has the same architecture as the generator. On other models, they get higher test accuracy than ResNet-18. Meanwhile, since OPS is a model-free method that takes advantage of the natural learning preference of neural networks, its transferability is better across different models. Besides different architectures, we also explore the impact on models with the same architecture but different capacities. We trained several WideResNets (Zagoruyko & Komodakis, 2016) with different sizes. The experiment results can be found in Table 3. From our observation, overparameterization, which is generally believed to enhance the ability to capture complicated features, does not circumvent the shortcut features.

Moreover, we observe that vision transformers are easily affected by manually crafted shortcuts, even though it is believed that their self-attention mechanism makes them less sensitive to data distribution shifts (Shao et al., 2021; Bhojanapalli et al., 2021). For CCT-7-3×1 and CVT-7-4 (Hassani et al., 2021), EM and OPS can degrade their test accuracy below 30% and 20%. This indicates that vision transformers may not generalize on out-of-distributions data as well as our expectations. If the training data is largely biased, *i.e.*, has notable shortcuts, and vision transformers will not perform much better than convolutional networks.

Table 5: Effectiveness of One-Pixel Shortcut & Error-Minimizing on ResNet-18 under different training strategies. Here $\ell_\infty$ AT stands for adversarial training with bound $8/255$.

| Training Strategy | Training Data | | | |
|---|---|---|---|---|
| | Clean | EM | OPS | CIFAR-10-S |
| Standard | 94.01 | 19.58 | 15.56 | 16.67 |
| Mixup | 94.75 | 38.18 | 33.13 | 23.23 |
| Cutout | 94.77 | 25.83 | 61.68 | 24.38 |
| RandAugment | 94.91 | 51.66 | 71.18 | 39.62 |
| RandAugment + Cutout | 94.86 | 36.26 | 79.70 | 33.85 |
| $\ell_\infty$ AT | 82.72 | 83.02 | 11.08 | 10.61 |
| $\ell_\infty$ AT + Mixup | 87.90 | 86.71 | 10.97 | 13.77 |
| $\ell_\infty$ AT + Cutout | 84.58 | 84.24 | 24.60 | 23.78 |
| $\ell_\infty$ AT + RandAugment | 85.50 | 85.06 | 44.86 | 46.23 |

## 4.3 Effectiveness under different training strategies

To evaluate the effectiveness of OPS under different training strategies, we train models on OPS perturbed data using adversarial training and different data augmentations such as Mixup (Zhang et al., 2018), Cutout (DeVries & Taylor, 2017) and RandAugment (Cubuk et al., 2020). Simple augmentations like random crop and flip are used by default in standard training. Models are also trained on EM perturbed data. As shown in Table 5, we can observe that both EM and OPS have a good performance on data protection, which degrade test accuracy to 19.58% and 15.56%. As mentioned in previous works (Huang et al., 2020a; Fu et al., 2021), EM can not work so effectively under adversarial training, and the model can reach an even higher accuracy than adversarially trained on clean data. Meanwhile, OPS can still keep effective under adversarial training. However, when it comes to data augmentation, EM seems more impervious, while OPS is more sensitive, especially to Cutout and RandAugment. This is due to the fact that EM injects global noises into images, while OPS only modifies a single pixel, which is equivalent to adding a very local perturbation. Adversarial training, which can be regarded as a kind of global augmentation, is able to attenuate the dependence on global shortcuts. On the other hand, local data augmentations like Cutout make models less sensitive to local shortcuts.

Naturally, for the purpose of complementing each other, we can combine EM and our proposed OPS together to craft a kind of ensemble shortcut. Since OPS only modified a single pixel, after being applied to EM perturbed images, the imperceptibility can still be guaranteed. We evaluate the effectiveness of this ensemble method under different training strategies and find that it can always

keep effective. Even if we use adversarial training and strong data augmentation like RandAugment, it is still able to degrade test accuracy to a relatively low level. Based on this property, we introduce CIFAR-10-S, where all the images are perturbed by the EM-OPS-composed noises. It can serve as a new benchmark to evaluate the ability to learn critical information under the disturbance of composed non-semantic representations.

We also extend our method to multi-pixel scenarios. According to Table 6, as the number of perturbed pixels increases, the test accuracy can be degraded to a lower level. Nevertheless, the more pixels are perturbed, the imperceptibility gets weaker, as illustrated in Figure 5. From our experiment on ResNet-18, 3-Pixel Shortcut can easily degrade the test accuracy to 9.74%. Moreover, more perturbed pixels alleviate the sensitivity to different data augmentations. For RandAugment, one more perturbed pixel can degrade the test accuracy to 46.45%, which is much lower than 71.18% of OPS.

Table 6: Effectiveness of Multi-Pixel Shortcut on ResNet-18 under different training strategies

| Training Strategy | Number of Perturbed Pixels | | | | |
|---|---|---|---|---|---|
| | 0 | 1 | 2 | 3 | 4 |
| Standard | 94.01 | 15.56 | 15.24 | 9.74 | 9.71 |
| Mixup | 94.75 | 33.13 | 9.99 | 9.94 | 10.02 |
| Cutout | 94.77 | 61.68 | 39.99 | 25.42 | 20.15 |
| RandAugment | 94.91 | 71.18 | 46.45 | 34.66 | 33.51 |

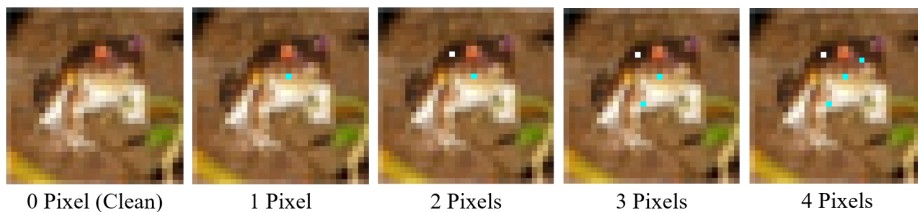

| 0 Pixel (Clean) | 1 Pixel | 2 Pixels | 3 Pixels | 4 Pixels |

Figure 5: Multi-Pixel Shortcut examples. More perturbed pixels lead to lower imperceptibility.

## 5 DISCUSSION AND CONCLUSION

In this paper, we study the mechanism of recently proposed unlearnable examples (ULEs) which use error-minimizing (EM) noises. We figure out that instead of semantic features contained by images themselves, features from EM noises are mainly learned by DNNs after training. These kinds of easy-to-learn representations work as shortcuts, which could naturally exist or be manually crafted. Since DNNs optimized by gradient descent always find the solution with the minimum norm, shortcuts take precedence over those semantic features during training.

We find that shortcuts can be as small as even a single pixel. Thus, we propose One-Pixel Shortcut (OPS), which is an imperceivable and effective data protection method. OPS does not require a generator model and therefore needs very little computational cost and has better transferability between different models. Besides, OPS is less sensitive to adversarial training compared to EM ULEs. We investigate the effectiveness of OPS and EM under different training strategies. We find that EM and OPS have their respective advantages and disadvantages. While EM cannot keep effective under global data augmentations like adversarial training, OPS is sensitive to local data augmentations like Cutout. Based on our investigation, we combine EM and OPS together to craft a kind of stronger unlearnable examples, which can still keep imperceptible but more impervious, and consequently introduce CIFAR-10-S, which can be a new benchmark. Besides, we have also discussed our method in multi-pixel scenarios.

There are still questions that need to be discussed in the future. Besides shortcuts that are crafted deliberately for the purpose of data protection, there are also shortcuts that naturally exist due to the inevitable bias during data collection. They can be the crux of network generalization on unseen data. How to identify and avoid them (*e.g.*, design data-dependent augmentation) is a challenging problem. We believe our work will shed light on the important impacts of shortcuts, and provide inspiration to harness them for more practical applications.

ACKNOWLEDGEMENT

This work is partly supported by National Natural Science Foundation of China (61977046, 61876107, U1803261), Shanghai Science and Technology Program (22511105600), and Shanghai Municipal Science and Technology Major Project (2021SHZDZX0102). Cihang Xie is supported by a gift from Open Philanthropy. In addition, we sincerely thank the reviewers for their valuable discussions.

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

# APPENDIX

## A  FURTHER INVESTIGATION ON SHORTCUT GENERATION

In Sec 3.2, we craft OPS by perturbing a fixed position to a fixed color target for each class. What if we only constrain one of these two properties? In other words, if perturbing a fixed position to different color targets, or perturbing different positions to a fixed color target, will a strong shortcut be created? With curiosity, we further explore these two properties individually.

We use OPS-Position and OPS-Color to represent the former and the latter setting. For OPS-Position, after finding the optimal position for each class via Algorithm 1, we perturb pixels at this position to random colors. For OPS-Color, we assign each class a predefined color (we add two colors since there are only 8 boundary colors) and perturb pixels at different positions to this color. Note that there might be many positions where the original color is already the same as the predefined color. We perturb the position where the original color is the furthest from the predefined color (measured by the $l_2$ norm) for each sample. As shown in Table A, the results indicate that if we only constrain one property, the resulting shortcuts will be harder to be captured. Besides, random perturbations at a fixed position can serve as a stronger shortcut than a fixed color at different positions.

Table A: One-Pixel Shortcut generated by different constraints. If perturbing a fixed position to different color targets or perturbing different positions to a fixed color target, the ability to degrade DNNs training will be less strong.

| Training Data | Clean | OPS | OPS-Position | OPS-Color |
|---|---|---|---|---|
| Clean Test Acc. | 94.01 | 15.56 | 46.22 | 85.34 |

## B  BASELINE OF A RANDOM PIXEL

For the purpose of evaluating our searching algorithm proposed in Sec. 3.2, we compare it with the baseline, which randomly chooses a fixed position and fixed color (denoted as RandPix) for each class to craft the shortcut training set, and report the clean accuracy of 10 runs in Table B. Since 15.56% is out of the three standard deviation intervals, our OPS design is validated to be meaningful.

Table B: Comparison with the randomly chosen perturbed position and target color.

| Training Data | Clean | OPS | RandPix |
|---|---|---|---|
| Clean Test Acc. | 94.01 | 15.56 | $50.67 \pm 11.29$ |

## C  SEARCHING OPS ON A SMALL PROPORTION OF DATA

In Algorithm 1, we search the entire dataset to generate OPS. What if the data is gradually added to the dataset? In other words, will the perturbed position and target color of OPS that are found on a small proportion of data serve as a strong shortcut on the entire dataset? Moreover, users might want to add their own noise before uploading their data to the dataset. From a practical standpoint, we further explore the potential of OPS by designing the following experiments.

Firstly, we take out different proportions of training data from the CIFAR-10 training set to search for the perturbed position and target color. Then we perturbed the whole training set based on the results we get from the selected small proportion of data, and train DNNs on the perturbed data. Secondly, considering the scenario that users add their own noise before uploading data to the dataset, based on the setting of the first experiment, we additionally inject random noise sampled from the uniform distribution $[-\epsilon, \epsilon]$ (here we set $\epsilon = 8/255$, following the common setting of adversarial learning) to the unselected data.

The results are shown in Table C, where we denote the first setting as *Clean-Upload* and the second setting as *Noisy-Upload*. Even if we only use $1\%$ of the whole dataset to search the perturbed position and target color, OPS is still able to degrade the clean accuracy of the trained DNN to $32.87\%$. Besides, additional noise does not degrade the efficiency of OPS. To some degree, the experiments prove the generalization capability and robustness of OPS from a practical standpoint.

Table C: We use different proportions of data to search the perturbed position and target color of OPS. Even if we only use $1\%$ of data, OPS can still largely degrade the clean accuracy of the trained network.

| Updating Type | Searching Proportion | | |
| --- | --- | --- | --- |
| | $1\%$ | $10\%$ | $100\%$ |
| Clean-Upload | 32.87 | 17.07 | 15.56 |
| Noisy-Upload | 32.10 | 17.30 | 15.56 |

## D EXPLORATIONS ON $\mathcal{L}_2$ ADVERSARIAL TRAINING

In Sec 4.3, we have studied the effectiveness of different types of shortcuts under different types of training strategies, including $l_\infty$ adversarial training and various data augmentations. For a more comprehensive exploration, we additionally evaluate them on $l_2$ adversarial training. Besides the commonly used setting ($\epsilon = 0.5$), we also try larger attack budgets, as shown in Table D.

Compared to $l_\infty$, $l_2$ is proved to be a more efficient AT manner for alleviating OPS. Nevertheless, when the perturbation budget is not large enough, the network will still be affected by the shortcuts. If we further enlarge $\epsilon$, it will hurt the clean test accuracy to a greater extent. As a local shortcut, OPS displays stronger inertia under $l_2$ AT, which can be viewed as a global data augmentation. As for EM, although generated by $l_\infty$ perturbations, is still sensitive to $l_2$ AT. These results conform to our discussion about the properties of local&global shortcuts and local&global data augmentations in Sec 4.3.

Table D: ResNet-18 trained on different training data using $l_2$ adversarial training with different attack budgets.

| Training Strategy | Training Data | | |
| --- | --- | --- | --- |
| | Clean | OPS | EM |
| Standard | 94.01 | 15.56 | 19.58 |
| $l_2$ AT ($\epsilon=0.5$) | 87.68 | 43.26 | 71.00 |
| $l_2$ AT ($\epsilon=1$) | 82.58 | 50.29 | 81.75 |
| $l_2$ AT ($\epsilon=2$) | 73.45 | 73.70 | 74.51 |

