# OpenReview forum: "One-Pixel Shortcut: On the Learning Preference of Deep Neural Networks"
_ICLR.cc/2023/Conference — ICLR 2023 notable top 25%_

### Official Review · Reviewer_qd1o · 2022-10-21

**Confidence:** 4
**Correctness:** 2
**Technical Novelty And Significance:** 2
**Empirical Novelty And Significance:** 2
**Recommendation:** 6

**Clarity, Quality, Novelty And Reproducibility:**

The paper is novel, clear and easy to read except for section 3.1. The paper gives implementation details and code so the paper can be reproduced.

**Strength And Weaknesses:**

Strengths:
1) The authors evaluate their method on several architectures.
2) The authors evaluate various training strategies as defenses against their proposed attack. They investigate data augmentations and adversarial training.

Weaknesses:
1) Subsection 3.1 is unclear and could be re-worked. Table 1 is not clear and could be better explained: the metric could be stated more clearly.
2)  In Table 5, CutOut and RandAugment seem individually quite effective against the one pixel shortcut method. This table is missing the combination of CutOut and RandAugment. It seems that such a combination could strongly affect the proposed one pixel shortcut method (as the individual components are already effective). Furthermore, modern architectures such as ViT use considerable amount of data augmentations so it makes sense to study combination of augmentations.
3) As easy missing training strategy in this study as mean of defense against the proposed one pixel shortcut, I would like to point to median filters and gaussian blurring.
4) Using l-inf perturbations for adversarial training seems unfair as by construction such perturbations will not be able to compensate the pixel replacement of the proposed OPS method. Using l2 perturbations with medium-strong perturbation radiuses seems much more appropriate and interesting in this setting. Especially as adversarial training with l-inf perturbations already lowers the clean test accuracy due to the robustness/accuracy tradeoff observed with adversarial training.

**Summary Of The Paper:**

This paper aims at preventing data from being used (without authorization) for training DNNs. They propose the One-Pixel Shortcut (OPS) method where all the images of a same class get the same pixel location replaced with a same color pixel. This method fools the network during training as the network will use this replaced color pixel shared within the class as a shortcut during training to learn the class instead of using actual discriminative features. The authors evaluate this method on CIFAR-10 on several architectures from ResNet to Vision Transformers.

**Summary Of The Review:**

The paper is interesting, easy to read and proposes an easy strategy to prevent the unauthorized use of images for training. My concern is that modern architectures such as ViT use considerable amount of data augmentations whereas here data augmentations are studied separately and even individually they seem to reduce the impact of the proposed method. Furthermore, the authors only study l-inf adversarial perturbations which by construction will fail against the proposed method. Studying l2 perturbations should be more relevant.

---

> ### Author Response · Authors · 2022-11-19
> **Response to Reviewer qd1o (Part 1)**
>
> We want to firstly thank you for the detailed comments. We address your concerns below:
>
> > Q1. Subsection 3.1 is unclear and could be re-worked. Table 1 is not clear and could be better explained: the metric could be stated more clearly.
>
> Thanks for your careful reading. we have reworked Sec 3.1. For the shuffling experiment, we first explain our goal to demonstrate that DNNs learn the shortcut instead of images if we train with data with shortcuts. Then we illustrate our design: denote $\hat{D}=$ { $(x_i + \delta_{i},  y_i)$ } as the perturbed training set, where $\delta_i$ is the perturbation associated with the $i$-th example, and after shuffling the training set becomes $\hat{D}^\prime =$ { $(x_j + \delta_{i},  y_i)$ } . By evaluating DNNs on the shuffled dataset, we can know whether they learn representations from image { $x_i$ } or perturbations { $\delta_{i}$ }. Finally, we analyze the results, which show that OPS is a more prominent shortcut in view of DNNs. We hope our reorganization addresses your confusion.
>
> > Q2. About the study on combinations of data augmentations.
>
> We really appreciate your concern, and we agree that studying the combination of augmentations will make our work more comprehensive. Actually, we have studied some combinations in Table 5 from our paper, viewing adversarial training (AT) as a kind of augmentation. Following your advice, we add the experimental results of RandAugment + Cutout in the following table. As discussed in Sec 4.3, OPS is a local perturbation and naturally sensitive to local augmentations. In addition to Cutout, there are also many kinds of local augmentation policies included in RandAugment. Combining them together indeed neutralizes OPS to a greater extent.
>
> | Training Strategy / Training Data | Clean | OPS | EM | CIFAR-10-S|
> |---|---|---|---|:---:|
> |Standard|94.01|15.56|19.58|16.67|
> |Cutout|94.75|33.13|38.18|23.23|
> |RandAugment|94.77|61.68|25.83|24.38|
> |RandAugment + Cutout|94.86|79.70|36.26|33.85|
>
> On the other hand, as shown in Table. 5 from our paper, if we combine L-inf AT (which is a global augmentation) with Cutout or RandAugment, OPS can even degrade the network to a lower extent than only applying RandAugment. Our results indicate that different kinds of augmentations perform differently against shortcuts, depending on their specific types. We have add this insightful combination study to our revised submission, and we will explore more conbinations between adversarial training and data augmentations in the future version.
>
> Our paper, from another angle, reveals the importance of data augmentations and proves why they work (because they alleviate shortcut learning to some extent), rather than only aiming to show the poisoning efficiency of OPS under augmentations. Due to the inevitable bias during data collection, shortcuts naturally exist in the datasets. We surprisingly discover that a single pixel, which is the smallest unit of a digital image, can degrade the training of deep neural networks so significantly.
>
> > Q3. About median filtering and Gaussian blurring
>
> We sincerely thank you for pointing out one direction for further explorations. We have done additional experiments on median filtering and Gaussian blurring, shown in the following table.
>
> | Training Strategy / Training Data | Clean | OPS | EM |
> |---|---|---|---|
> |Standard|94.01|15.56|19.58|16.67|
> |Median Filtering (window size=3)|85.07|83.50|25.07|
> |Median Filtering (window size=5)|63.02|62.58|24.21|
> |Gaussian Blurring (sigma=0.5)|91.68|23.67|24.26|
> |Gaussian Blurring (sigma=0.6)|88.73|23.91|25.33|
> |Gaussian Blurring (sigma=0.7)|81.15|20.15|24.44|
>
> For median filtering, it can effectively attenuate OPS. But at the same time, the performance of the network trained on unmodified clean examples will also be degraded. EM, as a global shortcut, is less sensitive to median filtering. This is due to the ability of median filters to remove dispersive outliers and the different properties of OPS and EM that the former is local while the latter is global.
>
> For the Gaussian blurring, we notice that it will only hardly alleviate OPS. Moreover, it also largely degrades the performance of the network trained on unmodified clean examples when the standard deviation increases.
> We should note that median filtering or Gaussian blurring, as an unattractive strategy that will hurt the normal training of neural networks, is not usually used in real-world scenarios.

---

> ### Author Response · Authors · 2022-11-19
> **Response to Reviewer qd1o (Part 2)**
>
> > Q4. About L-2 adversarial training.
>
> Thank you for your insightful idea. Additional experiments on L-2 adversarial training (denoted as L-2 AT) are represented in the following table. Besides the standard perturbation budget $\epsilon=0.5$, we follow your advice to use larger budgets $\epsilon=1$, and $\epsilon=2$. Compared to L-inf, L-2 is indeed proved to be a more efficient AT manner. Nevertheless, when the perturbation budget is not large enough, the network will still be affected by the shortcuts. If we further enlarge $\epsilon$, it will hurt the clean test accuracy to a greater extent. As a local shortcut, OPS displays stronger inertia under L-2 AT, which can be viewed as a global data augmentation. As for EM, although generated by L-inf perturbations, is still sensitive to L-2 AT. These results conform to our discussion about the properties of local\&global shortcuts and data augmentations in Sec 4.3. Besides, we have added this meaningful exploration to Appendix D in our revised submission.
>
> | Training Strategy / Training Data | Clean | OPS | EM |
> |---|---|---|---|
> |Standard|94.01|15.56|19.58|
> |L-2 AT (eps=0.5)|87.68|43.26|71.00|
> |L-2 AT (eps=1)|82.58|50.29|81.75|
> |L-2 AT (eps=2)|73.45|73.70|74.51|

---

> > ### Comment · Reviewer_qd1o · 2022-11-29
> > **Thank you for the rebuttal**
> >
> > I thank the authors for their very thorough rebuttal which clarified most of my interrogations. I would suggest to add the RandAugment + Cutout result and the median/gaussian filtering results to the appendix (not now as the revision period is closed but when possible) as limitations are also very interesting and give a better intuition of the method and its working mechanisms. I am upgrading my score as the paper is good discussion material in the field and raises interesting questions/limitations.

---

> > > ### Author Response · Authors · 2022-12-02
> > > **Thank you for the advice**
> > >
> > > We are really grateful for your valuable comments, which indeed help us improve the comprehensiveness of our work. We will follow your advice and add the median/gaussian filtering results to the appendix in the future version.
> > > As you said, OPS provides new insights into the characteristics of DNNs, although having some limitations. We are happy that you could accept the value of our work, and we want to thank you for the kind reconsideration of your score.

---

> ### Author Response · Authors · 2022-11-28
> **Looking forward to further discussion**
>
> Dear Reviewer,
>
> We are grateful for your valuable comments. In our response, for a more comprehensive study, we have included additional experiments on the combination of data augmentation, median and Gaussian filters, and L-2 adversarial training with large budgets. Moreover, we have further clarified some unclear expressions in our paper. We have incorporated all the changes in our revised manuscript for your kind consideration. And we hope that your concerns have been addressed.
>
> As you might know that we are inching closer to the reviewer final recommendation deadline. We would like to discuss with you in this period and are willing to provide more according to your feedback or further questions.
>
> If you feel happy with our response, please consider updating your score. Feel free to reach us out in case you need any clarification.
>
> Paper1277 Authors

---

### Official Review · Reviewer_bTQi · 2022-10-22

**Confidence:** 3
**Correctness:** 3
**Technical Novelty And Significance:** 3
**Empirical Novelty And Significance:** 3
**Recommendation:** 8

**Clarity, Quality, Novelty And Reproducibility:**

Paper is generally very clear, and I had no issues understanding the paper. One thing I didn't find very useful is Fig4. and its relevance in the overall theme of the paper. Method is proposed on publicly available datasets, and pseudocode/algorithm (Alg.1) is very clear too.

**Strength And Weaknesses:**

Strengths:
- The paper provides a new perspective on learning a per-class noise that minimizes test accuracy while achieving good training accuracy. Shortcuts like these are easy to pick up by a network, especially if they are consistent across images in the same class.
- Evaluation is very thorough, with different architectures showing that the method is not dependent or sensitive to architecture choices, augmentations, or training strategies.

Weaknesses:
- The method is rather complex in the sense that since neural networks are good at picking up shortcuts, it should be possible to perturb *any* pixel with *any* color consistently across all images in the same class - as long as the values of $\sigma_k, \xi_k$ are different for different k. This should be an initial baseline to compare against OPS which is obtained by minimizing Eqn (2).
- One major downside of using OPS is that regardless of the claims about the perceptibility of the augmented images, it is extremely easy for a human (the reviewer in this case) to see exactly which pixel is perturbed. Unlike the EM method which has the capability to learn a per-image noise, the OPS method performs a class-wise perturbation. This is problematic for few reasons. First, the easy perceptibility of the augmented pixel can lead to manual solutions to fix this problem (one way to combat OPS - if done across the dataset, is to just find the pixel such that the standard deviation of the pixel is 0 across the images of that class - and replace it with a mean of neighboring pixels, etc.) - and finding an analogous way to "undo the noise" in EM is not easy. Secondly, if part of the information is perturbed, visual inspection is enough to discard the perturbed data and then learn a network, but in EM, this is not possible.

There are no comments about the "generalization capability" of this noise location and color. How robust is this in settings where more data is gradually added to the dataset (like continual learning, etc.)? The OPS shortcut is learned on the entire dataset, which is not what is done from a practical standpoint (an end user would want to add their own noise before uploading a picture to the internet). This issue need not be solved in the paper, but should be addressed.

Moreover, it would be interesting to see how a method like Noise2Void [1] can be modified to use as an adversarial method to combat OPS - which performs selective masking on the image.

[1] https://arxiv.org/pdf/1811.10980.pdf


**Summary Of The Paper:**

The paper proposes a one-pixel shortcut (OPS) method to generate unlearnable examples that would render a trained model perform not better than a random network on test examples. This performs better than the existing unlearning example generation method by error minimization (EM) at different settings. EM and OPS are combined to create a dataset named CIFAR-10-S, which injects shortcuts in the dataset to evaluate different models' resistance to them.

**Summary Of The Review:**

The one-Pixel shortcut method provides a new way of generating unlearnable examples. The paper is well-motivated and well-explained. The experiments are satisfactory. Although I would still argue that from a practical standpoint, EM unlearnable examples generation is more usable, since its not easily detectable upon an initial inspection. From a practitioner's standpoint, seeing that training losses are reaching 0, while the model gets poor test error, they would inspect the dataset first. OPS-based augmentations are easier to detect and remedy, once detected.

However, I like the idea and think that this paper can lead the way for mitigating some of the shortcomings of the method.

---

> ### Author Response · Authors · 2022-11-19
> **Response to Reviewer bTQi (Part 1)**
>
> We are grateful for your valuable comments and your appreciation of our idea. We address your concerns below:
>
> > Q1. About an initial baseline
>
> We sincerely thank you for providing the insightful viewpoint. Regarding your concern, additional experiments are conducted to investigate the initial baseline of a random pixel. We randomly choose a fixed position and fixed color (denoted as RandPix) for each class to craft the shortcut training set, and report the clean accuracy of 10 runs in the table below. Since 15.56\% is out of the three standard deviation intervals, our OPS design is validated to be meaningful.
>
> |Training Data|Clean|OPS|RandPix|
> |---|---|---|---|
> |Clean Test Acc.| 94.01 | 15.56 | 50.67 $\pm$ 11.29 |
>
> > Q2. About the visual imperceptibility and removability
>
> We really appreciate your concerns. Firstly, OPS can indeed be removed by the prompt of standard deviation, but the prerequisite is that the user obtains the full knowledge of the dataset, including the specific threat and its working mechanism behind. This is not realistic in real-world scenarios. From the perspective of adversarial attack and defense, the defender usually does not obtain any knowledge of the specific threat. If fully aware of the threat of EM, we could also use various denoise-based purifiers (e.g. taking advantage of diffusion models or GANs) or adversarial training to neutralize it.
>
> Secondly, we generally use the L-p norm to constrain the imperceptibility. EM and OPS can be respectively viewed as L-inf and L-0 constrained perturbations. It is hard to say which type of perturbed example is more visually perceivable. Moreover, when it comes to larger-sized datasets like ImageNet, OPS, as an extremely local perturbation, will be more imperceptible, while EM, as a global perturbation, might be more visually perceivable. Actually, there are many attractive works on one-pixel or sparse attacks in the realm of adversarial learning [1,2,3] which significantly inspire researchers and contribute to the development of the AI security community.
>
> The purpose of our work is not to repudiate the value of EM. We just want to further explore how the non-semantic and minuscule shortcuts can degrade the training of DNNs to what extent. And surprisingly, a single pixel can be such a strong shortcut that significantly degrades the DNN training. In fact, OPS is not proposed as the counterpart of EM. They are both manually crafted shortcuts with different forms and should be complementary to each other. Either OPS (the local shortcut) or EM (the global shortcut) has its pros and cons, as we discuss in Sec 4.3. And naturally, they can be combined to complement each other, which is the motivation of the CIFAR-10-S introduced in Sec 4.3. The value of our work is not only introducing a new poisoning method but also providing new insights into the defect of DNNs.
>
> > Q3. There are no comments about the "generalization capability" of this noise location and color. How robust is this in settings where more data is gradually added to the dataset (like continual learning, etc.)? The OPS shortcut is learned on the entire dataset, which is not what is done from a practical standpoint (an end user would want to add their own noise before uploading a picture to the internet). This issue need not be solved in the paper, but should be addressed.
>
> We are grateful for you providing such an interesting idea, which inspires us to further explore the potential of OPS. For the purpose of investigating the generalization capability, we design the following experiment. Firstly, we take out different proportions of training data from the CIFAR-10 training set to search for the perturbed position and target color. Then we perturb the whole training set based on the results we get from the selected small proportion of data, and train DNNs on the perturbed data. The results are shown in the following table.
>
> |Searching proportion|1%|10%|100%|
> |---|---|---|---|
> |Clean Test Acc.| 32.87 | 17.07 | 15.56 |
>
> Moreover, considering that an end user would want to add their own noise before uploading a picture to the Internet, based on the experimental setting above, we additionally inject random noise sampled from the uniform distribution $[-\epsilon, \epsilon]$ (here we set $\epsilon = 8/255$, following the common setting of adversarial learning) to the unselected data. Even so, OPS generated on small-scale data also serves as a strong shortcut that largely degrades the training of DNNs.
>
> |Searching proportion|1%|10%|100%|
> |---|---|---|---|
> |Clean Test Acc.| 32.10 | 17.30 | 15.56 |
>
> To some degree, the experiments above prove the generalization capability and robustness from a practical standpoint. We will add this interesting and meaningful exploration to the future version of our paper.

---

> ### Author Response · Authors · 2022-11-19
> **Response to Reviewer bTQi (Part 2)**
>
> > Q4. Moreover, it would be interesting to see how a method like Noise2Void [4] can be modified to use as an adversarial method to combat OPS - which performs selective masking on the image.
>
> Thanks for your insightful idea. We carefully read the original paper [4], and find the official implementation based on the BSD setting where the image size is cropped to $256 \times 256$. Unfortunately, due to the time limitation, we are not able to adopt it to our setting yet. However, we want to admit that OPS will be degraded by median filtering, as shown in the following table. Since such a simple filter can attenuate OPS, we think the trainable DNN-based Noise2Void could naturally be an effective method. We will try to adopt it to our setting in the next weeks.
>
> | Training Strategy / Training Data | Clean | OPS | EM |
> |---|---|---|---|
> |Standard|94.01|15.56|19.58|
> |Median Filtering (window size=3)|85.07|83.50|25.07|
> |Median Filtering (window size=5)|63.02|62.58|24.21|
>
> In a word, since the generation of OPS is simple and straightforward, it would not require much effort to combat it, if knowing the details of its generation. The interesting part of OPS is that it reveals a bug or characteristic of general neural networks, as the reviewer vxV5 says. Before OPS is introduced, we are not likely to expect that a single pixel can have such a strong impact on the DNN training.
>
> > Q5. One thing I didn't find very useful is Fig4. and its relevance in the overall theme of the paper.
>
> Thanks for your question. Fig.4 shows that the data with shortcuts like EM or OPS can provide a optimum which is closer to the initialization. Denote $\theta$ and $\theta_0$ as the parameters of the network after training and at initialization. For instance, as we can see, for the first time when the training accuracy reaches as high as 90\%, the parameter change $\Vert \theta - \theta_0 \Vert_F$  of the network trained on data with shortcuts like EM or OPS is much less than that trained on data without shortcuts.
>
> According to the viewpoint of previous works [5, 6], DNNs are lazy and tend to find the solution close to the initialization if using gradient descent during training. We use Fig. 4 to help explain how shortcuts degrade the network training: by providing a solution which is closer to the initialization and enticing the network to find it.
>
> ### Reference
>
> [1] One pixel attack for fooling deep neural networks, IEEE TEC 2019.
>
> [2] Sparse and Imperceivable Adversarial Attacks, ICCV 2019.
>
> [3] GreedyFool: Distortion-Aware Sparse Adversarial Attack, NeurIPS 2020.
>
> [4] Noise2Void - Learning Denoising from Single Noisy Images, CVPR 2019.
>
> [5] The marginal value of adaptive gradient methods in machine learning, NeurIPS 2017.
>
> [6] Learning overparameterized neural networks via stochastic gradient descent on structured data, NeurIPS 2018.

---

> ### Author Response · Authors · 2022-11-28
> **Looking forward to further discussion**
>
> Dear Reviewer,
>
> We really appreciate your valuable comments. In our response, we have included additional experiments on the initial baseline with a randomly chosen location and target color, and further investigated the generalization capability of OPS from a practical standpoint. Moreover, we have further clarified some unclear expressions in our paper. We have incorporated all the changes in our revised manuscript for your kind consideration. And we hope that your concerns have been addressed.
>
> As you might know that we are inching closer to the reviewer final recommendation deadline. We would like to discuss with you in this period and are willing to provide more according to your feedback or further questions.
>
> If you feel happy with our response, please consider updating your score. Feel free to reach us out in case you need any clarification.
>
> Paper1277 Authors

---

### Official Review · Reviewer_vxV5 · 2022-10-28

**Confidence:** 3
**Correctness:** 4
**Technical Novelty And Significance:** 4
**Empirical Novelty And Significance:** 4
**Recommendation:** 8

**Clarity, Quality, Novelty And Reproducibility:**

Apart from a few issues surrounding some mathematical notations (please see the comments section), the paper is well written with motivations explained clearly.

To the best of my knowledge, the idea about the ability of perturbing one pixel to drastically change the behavior of a network is novel, and has been shown to be generalizable enough.

**Strength And Weaknesses:**

The method is well motivated, very simple, and most importantly, surprisingly effective at creating unlearnable examples.

The results are not simply empirical, but the authors also analyze what goes on with the created ULEs. For example, the authors visualize what happens to the feature computed through ULEs and show that with just one pixel change, all the intermediate features change their properties. This result will be surprising to the community. That being said, it also needs to be studied a bit more. In other words, in what way have the community understood the convolutional layers incorrectly which led to us assuming something like this (Fig. 1) happens.

The phenomenon discovered by the authors seems to generalize well enough; in that it not only affects the traditional convolutional architectures (e.g. ResNet) but also to transformer based architectures (ViT). This corroborates the fact that this result is indeed not an exploitation of a specific weakness, but is a bug (or feature) of neural networks in general.

Comments/Questions

“Images belonging to the same category are perturbed at the same position”. If I understand correctly, if you have two images of dogs, both will be perturbed at the same location with the same perturbation. How are these two properties related? In other words, what happens if you enforce one of the constraints (need for the perturbation to be in the same location) but not the other, and vice versa. This property should be studied better.

Mathematical notations are confusing sometimes. For example, in Eq. 1, it is not clear what c (in D_c, k) and i, j (in sigma_k(i, j)) mean.

Is there any trend in the discovered location for perturbation? In other words, is it the case that for certain classes, the perturbed locations turn out to be in center whereas for some other they are on the edge?


**Summary Of The Paper:**

The paper presents a method to generate unlearnable examples (ULE), using which if one trains a neural network, that network will not perform well (sometimes random performance) on an unclean test. The authors show that even changing just a small pixel value, given that that change is done consistently at the same location in all the images of the same class, can lead to surprisingly effective unlearnable examples. The application of such unlearnable examples is shown in many image classification settings (e.g., CIFAR-10) and superior performance over the existing paradigm of creating ULEs is shown.


**Summary Of The Review:**

The findings of the paper are surprising. The community of machine learning for privacy will definitely find the work useful. The method introduced is simple and effective for an important task. The authors have shown the generalization ability of the method through multiple lens (different tasks, different network architectures). I hence recommend acceptance.

---

> ### Author Response · Authors · 2022-11-19
> **Response to Reviewer vxV5**
>
> Thanks for your valuable comments and your appreciation of our findings. We address your concerns below:
>
> > Q1. Individually study on the position and the color
>
> Thank you for your insightful comments. It would be interesting and comprehensive to explore these two properties individually. The results are shown in the table below, where we denote OPS-position as only constraining the same position and OPS-color as only constraining the same color.
>
> For OPS-position, after finding the optimal position for each class via the searching algorithm, i.e. Algorithm 1 in our paper, we perturb pixels at this position to random colors. For OPS-color, we assign each class with a predefined color (we add two colors since there are only 8 boundary color). Note that there might be many positions where the original color is already the same as the predefined color. We perturb the position where the original color is farthest from the predefined color (measured by L-2 norm) for each sample.
>
> |Training Data|Clean|OPS|OPS-position|OPS-color|
> |---|---|---|---|---|
> |Clean Test Acc.| 94.01 | 15.56 | 46.22 | 85.34 |
>
> The results indicate that if we only constrain one property, the resulting shortcuts will be harder to be captured. Besides, random perturbations at a fixed position can serve as a stronger shortcut than a fixed color at different positions. We have added this interesting exploration in the appendix of our revised submission.
>
> > Q2. About confusing math notations.
>
> Thanks for your careful reading. The $c$ in $D_{c,k}$ stands for the word 'clean'. We denote the unmodified clean dataset as $D_{c}$ , and accordingly the clean subset of class $k$ as $D_{c,k}$ .
>
> Moreover, $\sigma_k \in \mathbb{R}^{H \times W}$ represents a position map of class $k$, where $H$ and $W$ are the height and width of every image. Consequently, $\sigma_k(i, j)$ means the element at the $i$-th row and the $j$-th column of $\sigma_k$ .
>
> Following your advice, we have clarified relevant notations in Sec. 3.2.
>
> > Q3. Is there a trend in the discovered location for perturbation ?
>
> Thanks for bringing out this interesting question. Since our calculations are purely based on data, there may be potential trend due to the bias of in-class data as you guess. For instance, airplane images usually come with blue sky, so the OPS pixel tend to be at the background on the edge. For some classes without a prominent foreground, the pixel would distribute randomly. In a word, it depends on the specific dataset we use.

---

> ### Author Response · Authors · 2022-11-28
> **Looking forward to further discussion**
>
> Dear Reviewer,
>
> We really appreciate your valuable comments. In our response, we have included additional experiments on the individual study of only location and color and clarified some confusing explanations. We have incorporated all the changes in our revised manuscript for your kind consideration. And we hope that your concerns have been addressed.
>
> As you might know that we are inching closer to the reviewer final recommendation deadline. We would like to discuss with you in this period and are willing to provide more according to your feedback or further questions.
>
> If you feel happy with our response, please consider updating your score. Feel free to reach us out in case you need any clarification.
>
> Paper1277 Authors

---

> ### Comment · Reviewer_vxV5 · 2022-12-02
> **Response to the authors**
>
> I thank the authors for their rebuttal. As for the first comment (studying the effects of the same color vs same position), the presented results seem to suggest that fixing the position of the perturbation (different color for different images) is a stronger constraint to reduce the classification error than fixing the color of perturbation (different position for different images). I think that this result can still be investigated further. For example, do we have the above results because of the following there is some position bias in the main object within a class (all dogs are on the left side), but that bias does not exist to the same extent for the color (all dogs are not of same color)?
>
> In any case, even in its current form, I think the paper is of good quality whose results will be useful to the community. I am thereby increasing my initial rating.

---

> > ### Author Response · Authors · 2022-12-05
> > **Thank you for the response**
> >
> > Thank you for the valuable discussion. We find that the fixed perturbed position is indeed a stronger constraint than the fixed target color. From our viewpoint, since there are various colors in a clean image, it is very likely that different positions in different images have already obtained the same color. And according to the training on clean images, which produces normally generalized model, the fixed color in different positions might not be a strong enough shortcut.
> >
> > However, the position bias of the main object within a class, as a naturally existing and semantic shortcut, might also be a possible factor. In addition to those non-semantic shortcuts like OPS, it will also be interesting to study the naturally existing and semantic ones, which might further help DNNs obtain better generalization. We are grateful for you pointing this out and will further investigate this question in the future.
> >
> > And finally, we sincerely appreciate the kind reconsideration of your score.

---

### Decision · Program_Chairs · 2023-01-20

**Decision:**

Accept: notable-top-25%

**Justification For Why Not Higher Score:**

The proposed approach is interesting, yet rather simple. The presented evaluation could also be stronger with respect to data augmentations.

**Justification For Why Not Lower Score:**

The proposed approach to generate unlearnable examples is interesting and simple and allows some interesting insights. The paper should be accepted for presentation at ICLR.

**Metareview: Summary, Strengths And Weaknesses:**

The paper presents a novel method to generate so called unlearnable examples (ULE), examples a  network is expected to perform poorly on, even when only lightly but consistently changing the image values.
The proposed method is well motivated, very simple. It is also very effective.

While yielding overall a convincing performance in the reported experiments, a remaining concern is that modern architectures such as ViT use considerable amount of data augmentations. In this paper,  data augmentations are studied separately.

Overall, the paper received strong reviews, ranging fro 6 to 8.


**Note From Pc:**

if the above contains the word "oral" or "spotlight" please see: "oral" presentation means -> notable-top-5% and "spotlight" means -> notable-top-25%. As stated in our emails, we are disassociating presentation type from AC recommendations